# Assessing the Nexus on Local Perspective: A Quali-Quantitative Framework for Water-Energy-Food Security Evaluation in Neglected Territories

**Rita de Cássia Santos de Souza [1], Ana Paula Dias Turetta [1,2,3,*], Michelle Bonatti [4] and Stefan Sieber [4]**

1   Program of Territorial Development and Public Policy, Federal Rural University of Rio de Janeiro, Seropedica 23897-000, Brazil; ecourban.rita@gmail.com
2   Brazilian Agricultural Research Corporation (EMBRAPA Soils), Rio de Janeiro 22460-000, Brazil
3   Sustainable Land Use in Developing Countries (SusLAND), Leibniz Centre for Agricultural Landscape Research (ZALF), Eberswalder Straße 84, 15374 Müncheberg, Germany
4   Department of Agricultural Economics, Faculty of Life Sciences Thaer-Institute, Humboldt-Universität zu Berlin, Unter den Linden 6, 10099 Berlin, Germany; michelle.bonatti@zalf.de (M.B.); stefan.sieber@zalf.de (S.S.)
*   Correspondence: ana.turetta@embrapa.br; Tel.: +55-21-2179-4579

**Abstract:** There is limited focus on the water-energy-food (WEF) nexus approach at the local scale, particularly considering the social contexts of neglected territories. To contribute to this debate, we propose a framework to address this topic in an area in Angra dos Reis, Rio de Janeiro State, Brazil, as a case study. A survey was conducted regarding residents' perceptions, choices, practices and access to water, energy, and food. The interviews highlighted important topics on the WEF nexus. As a result, a set of 12 indicators with data available on official datasets was proposed, reflecting residents' perceptions of WEF safety. For each indicator, thresholds were established based mainly on the goals defined by Brazilian policies across different spheres—federal, state, or municipal. Thus, each indicator is multi-scale and integrative, since it comprises a local perspective (from the interview results), an official perspective (from the official databases), and a threshold. The results showed problems with energy and water supply, and the presence of home cropping, contrasting with residents who do not have access to basic food and experience extreme poverty. All these elements drew attention to the possibility of applying the WEF nexus approach, searching for feasible solutions which can support better decisions and governance in neglected territories.

**Keywords:** water-energy-food nexus; urban planning; neglected territories; open datasets; public policies

## 1. Introduction

In the face of global climate change and urbanization, efforts to promote the rational use of resources and sustainable development in cities are urgent and necessary. Exponential population growth and industrialization resulting from global socioeconomic transformations are contributing to the expansion of cities that put pressure on natural resources, compromising both their availability and quality [1,2]. By 2030, two-thirds of the world's population will live in cities, with most of this population in urban slums or irregular settlements [3]. Often, these neglected territories comprise precarious constructions located in peripheral regions, lacking infrastructure and conditions of habitability, as they are both the cause and consequence of social and environmental problems [4]. Some factors that contribute to the origin of these territories include the dispute over access to urban land in an increasingly capitalist economy, which contributes to the concentration of private property, generating a frame of spatial segregation, marked by social and environmental inequalities [5].

This complex scenario is especially a risk related to water, energy, and food scarcity, all essential resources for the well-being of human societies and important to reduce poverty and facilitate sustainable development [1,6]. The study of connections between these elements through the WEF nexus approach contributes to explicit trade-offs between humans and the environment, revealing conflicts and synergies through an integrated and holistic perspective, assisting in decision making [7]. The basis to understand this approach consists of analyzing the balance between the use of environmental resources and socioeconomic factors, which are influenced by global factors like demographic changes, markets, prices, technologies, and climate change, among others [8,9].

However, due to the socio-economic and environmental vulnerabilities experienced by the neglected territories, it is even more challenging to develop and apply an inclusive analysis that allows an integrated evaluation, providing critical information for land use planning of these areas [10]. Dikeç [11] states that the distribution of resources, risks, and damages manifests on a spatial level, constituting what Walker [12] called "spaces of vulnerability" and "spaces of well-being", where the communities are exposed to vulnerable situations, such as floods, areas for waste disposal and lack of green spaces. This idea meets the concept of "environmental justice" that stems from the struggles of social movements for the right to territories and the resources necessary for their survival and dignity, against the unequal distribution of the burden from the current development model [13].

Several authors have advocated a more socio-political approach for WEF nexus studies, considering issues such as poverty reduction, distributive justice, and governance disputes [14–19]. Leese and Meisch [20] launch the challenge of using the WEF nexus approach to promote fair resource distribution and poverty alleviation. Albrecht et al. [18] emphasize the contribution of the social sciences to an innovative approach of the nexus assessment. Bouzarovski and Simcock [21] and Bouzarovski et al. [22] demonstrate the relationship between energy poverty and socio-spatial inequality (mentioning the concept of "energy justice"). These authors attribute these facts to the dramatic growth of peripheral areas, which was not accompanied by adequate infrastructure. Regarding peripheral areas, Garcia et al. [23] identified a low acquisition of fresh food by residents in a municipality of São Paulo, Brazil, due to insufficient supply and lack of variety. Aliaga et al. [24] argue that a social approach to food and nutrition security allows us to deal with issues such as poverty and inequality.

There is concern and criticism regarding the applicability of the nexus WEF approach, as the analysis should be as faithful as possible to the reality of the territory studied [25,26]. The local scale has been considered the most appropriate for urban planning since it sheds light on the social dynamics, which constitutes a territory of experience, with concrete problems [27,28]. However, Endo et al. [29] draw attention to the difficulty of local studies to allow correlation with the other scales of the territory (regional and global). So, more than proposing indicators at the local level, they must be applicable at temporal and spatial scales.

This is also a relevant argument, favorable for participatory methodologies that demonstrate to be helpful approaches to link theory and practice in policymaking, enabling a better comprehension of human relations with nature, including local knowledge [18,29]. Participatory methodologies are efficient to stimulate awareness of environmental issues, promoting a reliable reading of the territory, from a transdisciplinary perspective [28,30,31]. Understanding the local context is fundamental to the search for solutions in urban resilience, contributing to a planning system that stimulates popular participation in strategic decisions regarding the city [28,32].

Stakeholder engagement is essential to promote a mutual understanding of what can be considered "security" in the context, which indicators should be analyzed, and which actions need to be implemented [33], also contributing to a long-term impact on the WEF nexus evaluation through mechanisms that vary from questionnaires to forums and other platforms of participation [1,30]. The most appropriate process, however, should consider the time of research and resources available [1].

The use of indicators to assess sustainability is a very useful tool because it enables transforming a highly complex element into a technical, objective language, thus allowing monitoring and comparison at different levels [34]. However, most WEF nexus studies have focused on quantitative "top-down type" indicators (based on technical-scientific expertise), while participatory indicators of the "bottom-up type" are recommended for qualitative analyses of sustainability, especially those related to food and water security [24,35–37]. In this regard, participation can balance qualitative and quantitative approaches, building transdisciplinary interactions among stakeholders' perspectives [31,38,39].

Hence, we propose a WEF nexus assessment for neglected territories in urban areas, considering the residents' choices and perceptions. The residents' survey was correlated with data available on official datasets. It allowed us to present a low-cost and feasible methodology able to evaluate the status of the WEF nexus.

## 2. Materials and Methods

### 2.1. Study Area

The Angra dos Reis municipality, on the South Coast of Rio de Janeiro State, Brazil, comprises one of the largest remnants of the Atlantic Rainforest in Brazil. The entire region is a center of biological diversity and endemism, a hotspot of biodiversity [40]. In fact, the region is a crucially important Biodiversity Corridor recognized by UNESCO (United Nations Educational, Scientific and Cultural Organization World Heritage Committee) since 1991 [41,42]. More recently, in 2019, Ilha Grande (part of the Angra dos Reis territory) became a UNESCO World Heritage Site (United Nations Educational, Scientific and Cultural Organization World Heritage Committee) [42].

For many years, this municipality suffered intense socio-environmental conflicts originating from urbanization derived from huge national investments [43]. The changes in land use along with land speculation, all associated with exponential demographic growth, forced the low-income population to move to peripheral areas. This causes environmental degradation through either deforestation or the disposal of untreated sewage into water bodies [44,45].

The Bracuí region, located in this municipality, is the scene of an intense dynamic of occupation that comes, in many ways, from the appropriation of extensive coastal areas by real estate capital, invasions, construction in preserved areas, clandestine parceling, and mangrove landfill, where there is little provision of basic urban services. Furthermore, this region encompasses very important traditional territories, including the Guarani Sapukai Village and Quilombo Santa Rita do Bracuí (Figure 1).

Particularly in Itinga (Figure 2), which is our case study, conflicts over land use intensified in the 1990s, when villagers organized an association that struggled to gain recognition for, and guarantee, their rights. With urban growth and the construction of the BR-101 highway in the 1970s, a process of occupation of the locality began, which was not accompanied by improvements in urban infrastructure or other urban services.

A topographic assessment made in 2016 by the municipal government identified the existence of 2650 households with a population of approximately 10,000 people [46]. However, only 55% are permanent residents [46], with the remainder being vacationers who only spend a few days a year at the site. This context increases the complexity of governance in Itinga.

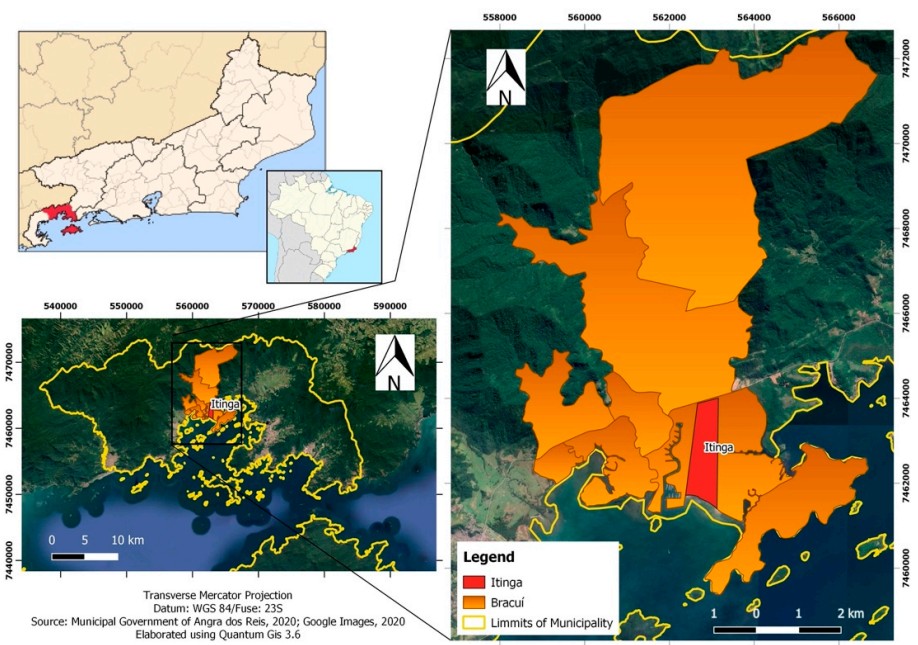

**Figure 1.** The location of Itinga, in Angra dos Reis, RJ. Source: Municipal Government of Angra dos Reis [46]. Elaborated by the authors using Quantum Gis 3.6.

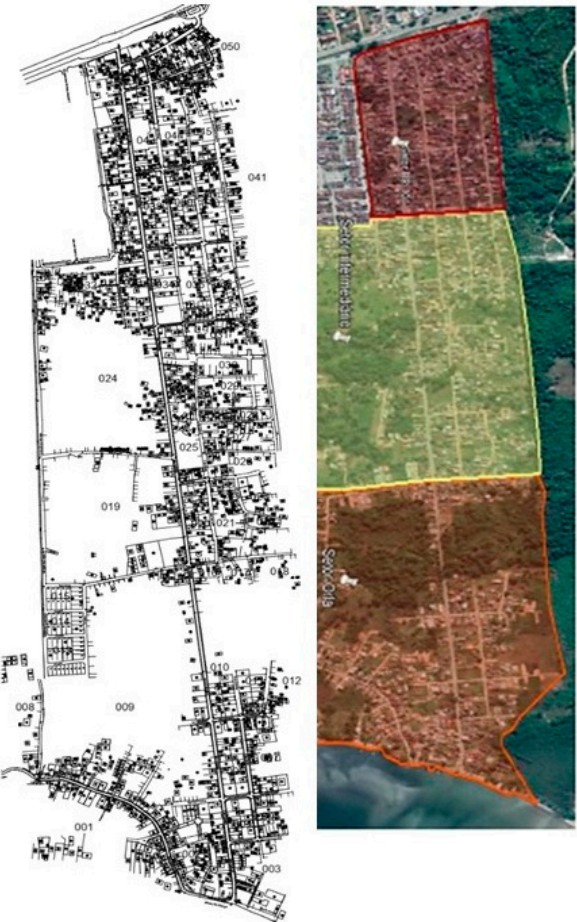

**Figure 2.** Planialtimetric map elaborated by municipality in 2016. In detail, the sectors I (red), II (yellow), and III (orange). Source: Municipal Government of Angra dos Reis [46]; Costa [47].

## 2.2. Data Collection and Sample

In 2018, anonymous semi-structured interviews were organized; the questionnaire was comprised of 17 questions about residents' perceptions, choices, practices, and access to resources such as fresh food, drinking water, and electricity (Table 1). In order to learn about the food items, number of meals per day, and seasonal variations in adult food intakes we used the 24-h recall method [48].

The sampling method used was the conglomerate. We split the neighborhood into three sectors, randomly selecting the blocks by drawing one at a time (Figure 2). If no respondent was available, the team picked the next household instead. We visited 22 households in total. All interviewees were over 18 years old and each interview, conducted on business days, lasted about 40 min. The Residents' Association was a support point for field activities, also helping to provide guidance on areas that could or could not be reached by the team, due to the high incidence of criminality and local conflicts (specifically bordering on the neighborhood and near the mangrove). All data were included in Microsoft Excel spreadsheets and analyzed using the Sphinx iQ2 software (test version).

## 2.3. Set of Indicators

The interview highlighted important topics regarding the WEF nexus from the community's perception. Then, we grouped these topics to subjects and establish the correspondence with data available on Brazilian data platforms that could work as indicators, considering the criteria of relevance—relevance of its production and use; validity—, indicator's ability to operationalize an abstract concept; reliability—quality of data collection; coverage—geographical and temporal scale; sensitivity—ability to reflect significant changes; specificity—ability to reflect changes strictly to the dimension of interest; and cost-effectiveness—cost and time to obtain adequate to the needs [49]. Only indicators related to the dimensions of access and availability at the household level were considered.

For water safety, we considered aspects related to basic human needs, such as water availability (including the diversity of water resources) and water quality, considered by Aboelnga et al. [33] as the main components of urban water security. For energy security, we considered criteria of availability and diversity, as presented by Azzuni and Breyer [29]. For food security, in turn, the home level is related to care, and we considered the aspects related to feeding practices, also referring to the socio-economic aspects of food security of families, the knowledge, habits, and decisions about what foods should be purchased, prepared, and consumed [50].

The number of indicators was defined in order to allow a quick qualitative and quantitative analysis, covering the different dimensions of securities at a local scale.

## 2.4. Definition of Thresholds

The third and last step was the definition of thresholds for each indicator. The thresholds were established mainly based on the goals defined by Brazilian legislation and policies across different spheres—federal, state, or municipal. For those indicators that did not have specific targets on national bases, we followed the recommendations of international organizations, such as the United Nations (UN) or the World Health Organization (WHO).

The methodology is summarized in Figure 3.

**Table 1.** Indicator framework for water-energy-food nexus assessment at a local scale.

| Question | Indicator | Objective | Justification | Reference |
|---|---|---|---|---|
| **1. (a) How many days a week do you usually eat fruits?**<br>( ) 5 or more days<br>( ) 3 to 4 days<br>( ) 1 to 2 days<br>( ) rarely<br>( ) never<br>**(b) On an average day, how many times do you eat fruit**<br>( ) 1 time<br>( ) 2 times<br>( ) 3 or more times<br>**(c) How many days a week do you usually eat at least one type of vegetables (lettuce, tomato, cabbage, etc.—not potatoes, cassava, or yam)?**<br>( ) 5 or more days<br>( ) 3 to 4 days<br>( ) 1 to 2 days<br>( ) rarely<br>( ) never<br>**(d) On an average day, how many times do you eat vegetables?**<br>( ) 1 time<br>( ) 2 times<br>**(e) How many days a week do you usually drink natural fruit juice?**<br>( ) 5 or more days<br>( ) 3 to 4 days<br>( ) 1 to 2 days<br>( ) rarely<br>( ) never<br>**(f) On an average day, how many glasses of natural fruit juice do you drink?**<br>( ) 1<br>( ) 2<br>( ) 3 or more | 1F. Recommended consumption of FV | Quantify in percentage (%) recommended consumption of fruits and vegetables (FV). Considers the percentage of individuals who consume fruits and vegetables five or more times/day, five or more days/week (including consumption of natural juices), or on five or more days of the week (including the consumption of natural fruit juices). The calculation of the daily total portions is made considering each fruit or each fruit juice as equivalent to a portion. | It is recommended that fruits and vegetables be consumed five or more times per day on five or more days of the week (according to the World Health Organization). The recommended consumption of fruits and vegetables is associated with disease reduction and may also be correlated with social and economic issues, in addition to habits, according to some studies. | IBGE—National Survey of Health (PNS) |
| **2. How many days a week do you usually consume fish or seafood?**<br>( ) 5 or more days<br>( ) 3 to 4 days<br>( ) 1 to 2 days<br>( ) rarely<br>( ) never | 2F. Recommended consumption of fish/seafood | Quantify in percentage (%) consumption of fish and seafood at least once a week. | The American Heart Association (AHA) recommends, for individuals without diagnosed cardiovascular diseases, fish intake twice a week, with portions of 112 g each, especially those rich in fatty acids. The Brazilian Society of Cardiology (SBC) corroborates the AHA recommendation. However, the National Health Survey considers it appropriate to consume fish at least once a week. | IBGE—National Survey of Health (PNS) |

**Table 1.** *Cont*.

| Question | Indicator | Objective | Justification | Reference |
|---|---|---|---|---|
| **3. How many days a week do you usually eat beans?**<br>( ) 5 or more days<br>( ) 3 to 4 days<br>( ) 1 to 2 days<br>( ) rarely<br>( ) never | 3F. Regular consumption of beans | Quantify in percentage (%) of individuals who claim to consume beans on five or more days a week. | Recommendations of a healthy diet for the Brazilian population are consolidated in the Food Guide for the Brazilian Population, which proposes the daily consumption of a portion of beans. When combined with rice, it is complete in terms of protein. In addition, they are two sources of carbohydrates, giving energy and satiety, while simultaneously adding the benefits of minerals and some vitamins, mainly the B complex. | IBGE—National Survey of Health (PNS) |
| **4. Where do you usually buy fruits and vegetables?**<br>( ) in the neighborhood<br>( ) self-production<br>( ) farmer's market<br>( ) street vendor<br>( ) grocery store<br>( ) supermarket<br>( ) other neighborhood<br>( ) donations<br>( ) I don't know<br>( ) I don't consume these items | 4F. Local acquisition of FLV | Evaluate the local sources of fruits and vegetables by the community, (including local production). Expresses the percentage (%) of respondents who claim to purchase fruits and vegetables in local establishments (in the neighborhood). | Some studies indicate the influence of the location of acquisition establishments on regular consumption of fruits and vegetables. Thus, this indicator is considered in order to identify the main local sources of fruits and vegetables (and which are the most accessible), as well as the presence, or not, of local production. | IBGE—Consumer Expenditure Survey (POF) |
| **5. What is the origin of electricity in your residence?**<br>( ) General grid<br>( ) Diesel<br>( ) Solar<br>( ) Wind<br>( ) Hydropower<br>( ) Natural gas<br>( ) Kitchen stove<br>( ) Ethanol<br>( ) Mixed system<br>( ) Other source. What? ______<br>( ) None | 1E. Access to electricity | Aims to quantify the percentage (%) of the population that has regular access to electricity in their homes. | Among the Sustainable Development Goals defined by the United Nations, there is the universal, reliable, modern, and affordable access to energy services. Considering the areas of irregular occupation, energy supply becomes an emblematic issue. Some residents remain without regular access, with clandestine connections. | IBGE—Continuous National Household Sample Survey (PNAD) |
| **6. What kind of energy do you use to prepare food in your home?**<br>( ) electric<br>( ) natural gas<br>( ) kitchen stove<br>( ) wood/coal<br>( ) other. What? ________<br>( ) none | 2E. LPG (Liquefied Petroleum Gas) consumption in food preparation | Aims to quantify in percentage (%) of interviewees whose use LPG in food preparation not concomitantly with biomass sources. | Residential LPG consumption in Brazil is predominantly focused on food cooking. A considerable portion of LPG consumption is used for cooking by low-income families, with the costs for LPG making up a significant portion of family expenditures. | IBGE—Continuous National Household Sample Survey (PNAD) |

**Table 1.** *Cont.*

| Question | Indicator | Objective | Justification | Reference |
|---|---|---|---|---|
| 7. **How many times have you had no light in your house in the last year?** _______________ | 3E. Equivalent Frequency of Interruption | Expresses the number of interruptions that, on average, each consumer in the sample analyzed has suffered in the last year. | To evaluate the quality of public electricity distribution services, indicators of collective continuity are used. One of these indicators is the Equivalent Frequency of Interruption per Unit Consumer, expressed in number of interruptions and hundredths of the number of interruptions. The Brazilian National Agency of Electric Energy—ANEEL requires concessionaires to maintain a continuity standard and, for this, it edits limits for these indicators. | Brazilian Electricity Regulatory Agency- ANEEL |
| 8. **What was the maximum time the energy was out at your home in the last year?** _______________ | 4E. Equivalent Duration of Interruption | Expresses the average duration of interruptions reported by the interviewed residents (hours). | Another indicator of the quality of public electricity distribution is the Equivalent Duration of Interruption per Consumer Unit, expressed in hours and hundredths of hours. | Brazilian Electricity Regulatory Agency |
| 9. **Where does the water you use in your house come from?**<br>( ) public service<br>( ) artesian well<br>( ) natural spring (less than 30' of distance)<br>( ) natural spring (more than 30' of distance)<br>( ) mineral<br>( ) tank truck<br>( ) rain<br>( ) others. What? _____________<br>( ) none | 1W. Access to Water supply | Aims to quantify the percentage (%) of interviewees whose have water supply (either from the grid or other alternative source). | Regarding universal and equitable access to water, water supply must be ensured to everyone, regardless of social, economic, cultural, gender, or ethnicity conditions. This concept is aligned with the premise of access to water as an essential human right. Thus, it is important to monitor the deficit of supply to the population, according to different income strata. According to United Nations guidelines, the proportion of the population that has access to an improved source of water located on or near the property should be included, which is accessible with at most a 30-min round trip. Improved sources include water supply at home or property through the general network, as well as other forms of supply. | National Sanitation Information System (SNIS) |
| 10. **Do you notice changes in public water supply throughout the year?**<br>( ) lack of water mainly during high season<br>( ) lack of water mainly when it rains too much<br>( ) frequent lack of water, at any time of the year<br>( ) rarely lacks water<br>( ) never lacks water<br>( ) I am not supplied by the public network | 2W. Availability of public water supply services | Aims to quantify the intermittency or lack of water for the community. It considers the percentage (%) of individuals who claim no lack of water in the community. | Having access to the public water service does not necessarily mean that water is always available to users. It is known that the intermittency in supply and even the lack of water distribution is a reality for a significant part of cities. According to United Nation guidelines, water should be available whenever necessary. | National Sanitation Information System (SNIS) |
| 11. **Do you consider the water supplied by the public network:**<br>( ) good<br>( ) bad | 3W. Water quality | Aims to evaluate the quality of drinking water supplied publicly. We considered the percentage (%) of individuals who classify water as suitable for consumption. | According to United Nations recommendations, to universalize the access of water, water should be free of contaminants—suitable for human and animal consumption. This indicator is based on the perception of residents. It complements, therefore, the other indicators and is related to the degree of insecurity/safety of residents for water consumption of the public network. | Water Quality Surveillance Information System for Human Consumption (SISAGUA) |

**Table 1.** *Cont.*

| Question | Indicator | Objective | Justification | Reference |
|---|---|---|---|---|
| **12. What is the destination of the sewer of your house?**<br>( ) Public network<br>( ) Septic tank connected to the net<br>( ) Tank not connected to the net<br>( ) Ditch<br>( ) Directly in the river, lake or sea<br>( ) I don't know<br>( ) other. What? _______________ | 4W. Sanitary sewage | It aims to quantify the portion of the community that uses safely managed sewage treatment services. This indicator is calculated considering the percentage (%) of individuals who claim to have sewage connected to the public collection system, including septic tanks connected to the network. | Several studies reveal the association between the absence of sewage treatment and high rates of hospital admissions, proliferation of waterborne diseases, and high mortality rates, especially among infants. Interventions in basic sewage treatment directly reflect in the improvement of public health conditions, reducing the incidence of waterborne diseases. The effects of environmental degradation resulting from the lack of adequate collection and treatment of domestic sewage are also widely known. | National Sanitation Information System (SNIS) |

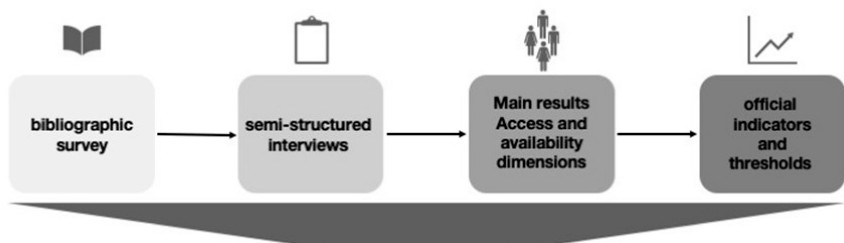

**Figure 3.** Construction of participatory indicators for nexus water-food-energy assessment in urban and peri-urban areas. Source: The authors.

### 3. Results and Discussions

A questionnaire comprising 17 questions was applied to Itinga's residents, following the sampling method explained in Section 2.2. The goal for each question is explained in Table 1.

Half of the interviewees were fifty years old or older, with 8 of them residing in the neighborhood for 6 to 10 years. Females comprised 13 of the interviewees; 8 of all interviewees were housewives. Most respondents (15) have a family monthly income of less than one minimum Brazilian wage per capita, equivalent to US$243.82.

Regarding water security aspects, 17 of the interviewees claimed that Itinga has a high natural availability of water. Indeed, the Municipal Basic Sanitation Plan [51] suggests the region has potential for exporting water, considered as the second major hydrographic basin in the municipality. Despite this, 14 of the interviewees report daily rationing of public water supply services. As a way to save water, 13 residents reported practices such as reusing greywater and rainwater harvesting (Figure 4).

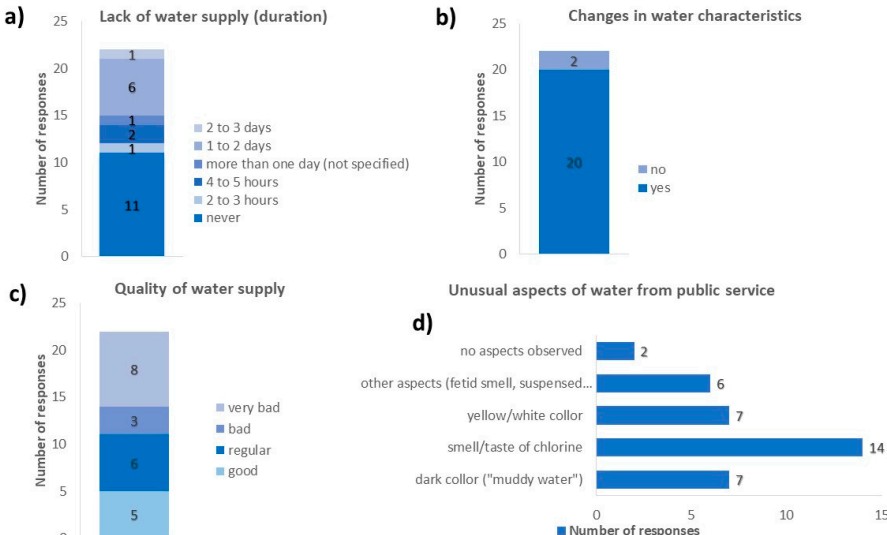

**Figure 4.** Main interviewees' results regarding water availability and access in Itinga, Angra dos Reis—Brazil. Source: The authors.

Almost all interviewees in this survey (21 respondents) have publicly supplied water. However, 8 of them consider the publicly provided water quality as bad while 6 respondents considered it regular in quality. Constant changes to water characteristics were noted by 19 of the interviewees. The main perceived changes were a strong smell, chlorine flavor, and a dark color ("muddy water"). This is the main reason they obtain water from a local spring for consumption.

According to the National Information System on Sanitation [52], 45% of the population in Angra dos Reis have access to sewage treatment. In Itinga, there is no public sewage treatment, depending exclusively on individual solutions. 17 of the interviewees affirmed using a septic tank and 13 respondents consider sewage treatment inadequate. The main problems related to sewage in the community are overflow and bad smells on rainy days or during high tide. This is a reality for many parts of the world. The World Health Organization [53] reports that at least 2 billion people around the world use a drinking water source that is contaminated with feces. Contaminated drinking water is estimated to cause 485,000 diarrheal deaths annually. Furthermore, by 2025, half of the world's population will live in water-stressed areas.

Half of the interviewees (11 of the responses) never had a problem with their water supply. For those experiencing an interruption in water supply, the maximum period without water was three days, affecting mainly the preparation of food. However, 3 interviewees said they needed to use a pumping system in the last three months, using an artesian well, with increases in energy expenses.

In Itinga, all sampled residents demonstrated access to the electrical grid, although it is known that, in 2016, about 80 families did not have regular electricity service because they lived in prohibited areas, without authorization (personal information). All interviewees complained about constant power outages, lasting up to three days. According to Almeida et al. [54], households living in peripheral communities of Latin American cities are subject to energy poverty, even if they have access to the electricity grid. Among the main causes are clandestine connections and criminal domains, which make it difficult for energy concessionaires to maintain facilities, causing frequent and lasting service interruptions. The low quality of electrical supply directly affects the comfort of low-income consumers. Frequent and long blackouts compromise thermal comfort and food storage, making it impossible for other domestic activities [55] (Figure 5).

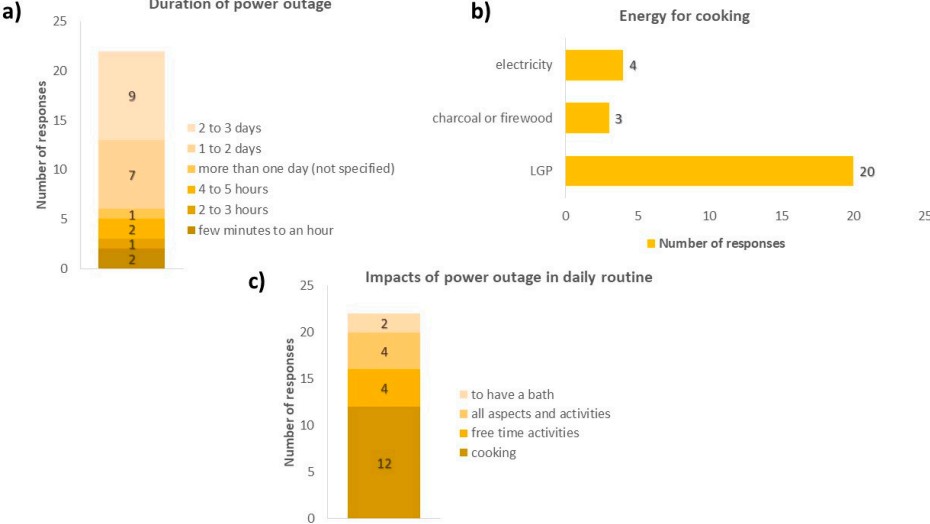

**Figure 5.** Main interviewees' results regarding energy availability and access in Itinga, Angra dos Reis—Brazil. Source: The authors.

Regarding food security, our results demonstrate low consumption of fresh food. Only one person consumes five or more daily portions of fruits and vegetables, while most consume only 1 to 2 portions (6 of the responses). The main reason for non-consumption or low consumption of fresh foods is economic issues. Fresh food is mostly obtained in the neighborhood itself, although 4 of those interviewed stated that they obtain fresh food through their own or family production. However, there are still families with low access to food in neighborhoods. This could be a consequence of the peri urbanizing phenomenon, as demonstrated by Garcia et al. [23]. The authors identify that the low acquisition of fresh

food by residents in peripheral areas of the São Paulo municipality, Brazil, is a consequence of insufficient supply and a limited variety of food. In this sense, it is possible to detect a link between peri urbanizing processes and food security (Figure 6).

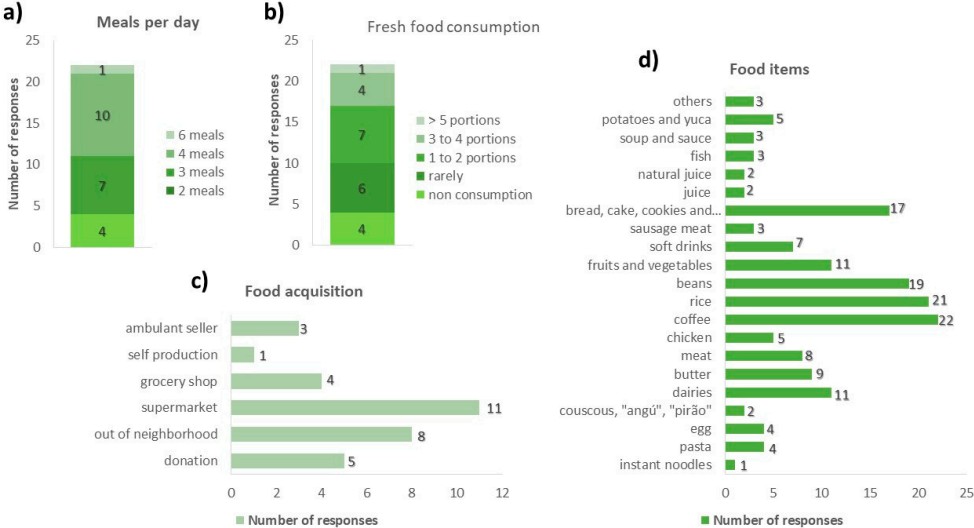

**Figure 6.** Main interviewees' results regarding food availability and access in Itinga, Angra dos Reis—Brazil. Source: The authors.

Organic foods are consumed by half of the respondents, mainly sourced from their own or family production. However, 2 respondents stated categorically that they were unaware of the concept of organic food. The consumption of local food plants was reported by 11 respondents. Taioba (Xanthoroma sagittifolium), a species considered to be part of the PANC (Edible Weeds) group, is the most popular. A study conducted by Almeida [55] shows that residents of peripheral urban communities in a southeastern Brazilian city, Belo Horizonte, use food self-supply based on local production. In addition to these crops, the project identified local knowledge about the use of native plants as a food source. Edible weeds emerged as an important solution that promotes quality food in the short term, especially among low-income families.

When asked about the possibilities for encouraging urban agriculture in the neighborhood, 10 interviewees supported the implementation of home gardens. Only 5 showed interest in community gardens, claiming the difficulty of integration among villagers. Ochoa et al. [56], studying urban gardens in different countries, find difficulties for community gardens during the formation phase and community building, when citizens work together to create a group of gardeners who participate in the gardening project. The same authors also state that top-down projects struggle with the financial resources needed to maintain gardens. With these barriers to overcome, case studies also show that active leadership is crucial for maintaining the activity.

Conceptual gaps were evidenced during the interviews, such as the knowledge about organic food and alternative energy sources. Interviewees showed that they did not have a formed opinion on these topics. Other topics, however, showed strong relevance among residents, especially water quality, due to uniformity in the pattern of responses. The use of the 24-h dietary recall method to characterize food consumption had an elevated rate of response rejection, especially in the most deprived families. On the other hand, the lack of integration among residents was highlighted as a main obstacle for developing sustainability actions, such as community gardens, for example.

Based on the questionnaire responses, we moved forward to the next step, which was to establish a correspondence between the questions (Table 1) and the data available in the official Brazilian database (Table 2).

The main public open datasets of Brazilian institutions were analyzed considering especially the geographical and temporal scale of the data available. Thus, we defined a set of 12 indicators that reflect residents' perceptions of WEF safety (Table 2). Additionally, for each indicator, a threshold was defined, based on existing public policies and programs in Brazilian legislation or, if no thresholds were established for the indicator, we considered international standards as a reference.

The thresholds are a fundamental aspect in sustainability assessments since they work as a reference of acceptable conditions within which we actively manage to maintain the status of the indicator [57].

**Table 2.** Thresholds for each water-energy-food security indicator.

| Water | | | Energy | | | Food | | |
| --- | --- | --- | --- | --- | --- | --- | --- | --- |
| Indicator | Threshold | Reference | Indicator | Threshold | Reference | Indicator | Threshold | Reference |
| 1W Water Supply | 100% | 2030 Agenda [58] | 1E Access to electricity | 100% | Sustainable Energy for All [59] | 1F Adequate consumption of FV | 43% | National Food and Nutrition Security Plan [60] |
| 2W Availability of Public Water Supply Services | 73.3% | National Basic Sanitation Plan [61] | 2E LPG consumption | 100% | Sustainable Energy for All [59] | 2F Recommended consumption of fish/seafood | 2 times week | World Health Organization [62] |
| 3W Quality of Public Water Supply Services | 100% | 2030 Agenda [58] | 3E Equivalent Frequency of Interruption | 9 times | Brazilian Electric Regulatory Agency [63] | 3F Recommended Bean Consumption | More than 5 times week | Food Guide for the Brazilian Population [64] |
| 4W Sewage Treatment | 100% | National Basic Sanitation Plan [61] | 4E Equivalent Interruption Duration | 11 h | Brazilian Electric Regulatory Agency [63] | 4F Local Acquisition of FV | 100% | National Food and Nutrition Security Plan [60] |

Therefore, each indicator is multi-scale and integrative, since it comprises a local perspective (from the interview results), an official perspective (obtained in the official databases, on a regional or national scale), and a threshold.

## 4. Conclusions

Faced with the challenges of climate change and the global phenomenon of increasing urbanization, rationalization of resources and the sustainable development of cities are increasingly important. Water, energy, and food are fundamental resources for human well-being. However, when they are distributed unequally across cities, it can result in conditions of water stress, energy poverty and food deserts.

In our study we presented a case study about a typical neglected neighborhood in Brazil. The questionnaire highlighted the residents' perception about the WEF nexus. Most of the residents reported a lack in accessibility and availability of the WEF services. It was also possible to realize the existence of conceptual gaps regarding some topis, e.g., organic food and alternative energy sources. The lack of community integration was also identified has an issue that made the development of integrative solutions difficult for the neighborhood.

The correlation of the questionnaire with data available on open official datasets allows an easy and low cost WEF nexus assessment and can be a useful tool for decision-making in neglected territories. It also highlights that regular data collection by the government across scales is essential for the best decisions to be made to reduce social inequalities and improve human well-being.

Our study contributes to the translation of household perceptions to eligible indicators, enriches sustainability analysis, and corroborates the transdisciplinary tendencies of the nexus approach. Although many technical sustainability indicators already exist, their effective implementation tends to be scarce because they are developed without stakeholder engagement, imposed through top-down approaches. This work also presents an approach that is simultaneously integrated, practical, and transdisciplinary, thus enabling the crossing

of information at different scales of the territory. The results clearly show the influence of urbanization processes on the threats facing the supply of basic services in neglected territories common in Latin American countries. It represents a low-cost framework that can be used by decision-makers without needing any specific training. Moreover, these are indicators that are internationally recognized, offering the possibility to be applied in multiple countries. In this way, it is also possible to monitor SDG achievements through the same indicators.

**Author Contributions:** Conception and design, R.d.C.S.d.S., A.P.D.T. and M.B.; writing, R.d.C.S.d.S., A.P.D.T., M.B.; editing, S.S. All authors have read and agreed to the published version of the manuscript.

**Funding:** This research is funded by the National Council for Scientific and Technological Development (CNPq)—441313/2017-5.

**Informed Consent Statement:** Verbal informed consent was obtained from each member of the sampled population prior to their interview. The participants were informed that the personal information provided was to remain confidential and only used for research purposes.

**Data Availability Statement:** All the data are available already in the manuscript.

**Acknowledgments:** We would like to thank the reviewers for their constructive comments, and to Itinga's residents for their availability to contribute with this study.

**Conflicts of Interest:** The authors declare no conflict of interest.

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
