# Peer review of "Assessing the Nexus on Local Perspective: A Quali-Quantitative Framework for Water-Energy-Food Security Evaluation in Neglected Territories"

_water, doi:10.3390/w14050731_

Round 1
Reviewer 1 Report
Manuscript no.: water-1583330-peer-review-v1
First author: Rita de Cássia Santos de Souza
Title of paper: Assessing the nexus on local perspective: a quali-quantitative 2 framework for water-energy-food security evaluation in ne-3 glected territories
I found this study interesting. In this manuscript, the authors propose a framework to address this issue considering an area in Angra dos Reis, Rio de Janeiro State, Brazil, as a case study. The authors should state more clearly the reason for doing the work for the international scientific community. The manuscript requires modifications and explanations before it is suitable for publication (see specific comments).
Specific comments
Title
please rewrite a shorter title.
Abstract
I suggest the authors to write clear the nature of the hypothesis and aims, major findings and conclusions.
Keywords
Replace two or more appropriate key words
Introduction
The Introduction should state more clearly the reason for doing the work with the nature of the hypothesis and the essential background. The text is long?
Materials and Methods
Generally, the text in each subchapter is difficult understandable from the readers.
Results and Discussion???
I am lost! The writing style of this section is insufficient and the text is long. Please discuss only the main results using recent references!
Conclusions
Write only the main conclusions.
Author Response
Dear Reviewer,
We hereby submit the revised version of the manuscript water-1583330 entitled “Assessing the nexus on local perspective: a quali-quantitative framework for water-energy-food security evaluation in neglected territories”.
After receiving your comments, we thank you very much for providing us the opportunity to further improve our manuscript. We appreciate your constructive criticism and have addressed the major concerns.
In the revised version, all comments were carefully addressed in the text and explained below. If additional changes are needed, we will be happy consider your recommendations. We appreciate your time and look forward to your response.
Sincerely yours,
The Authors
Specific comments
Title
please rewrite a shorter title.
We would like to maintain the original title, especially because the other reviewers didn’t request to change it.
Abstract
I suggest the authors to write clear the nature of the hypothesis and aims, major findings and conclusions.
We rewrote the abstract.
Keywords
Replace two or more appropriate key words
We updated the keywords.
Introduction
The Introduction should state more clearly the reason for doing the work with the nature of the hypothesis and the essential background. The text is long?
We believe that the introduction presents a coherent storyline. However, we tried to reduce it and make it clearer.
Materials and Methods
Generally, the text in each subchapter is difficult understandable from the readers.
We reviewed this section and rewrote some parts.
Results and Discussion???
I am lost! The writing style of this section is insufficient and the text is long. Please discuss only the main results using recent references!
We rewrote this section.
Conclusions
Write only the main conclusions.
We rewrote this section.

Reviewer 2 Report
The objectives of the manuscript (research) should be clearly identified, preferably at the end of Introduction.
L189: those seven criteria should be briefly explained.
Access is mentioned in the abstract, L190, Figure 3, Figure 4, etc. But it seems that for water, there is no indicator about access (L193). Probably, it is defined that if water is available to people, then it must be accessible. In other words, you considered that it does not make sense to say that water is available, and then state that it is not accessible. This seems logical, but please make it explicit. On the other hand, Access shows up in Table 1. So, things should be streamlined.
L207: WHO is an agency of UN, so you shouldn’t write “UN or WHO”. Besides, these are huge organizations. Please make their relevant publications or webpages very clear. You can make a list. Give explicit references and not huge organisations.
Table 2: Availability of Public Water Supply Services (2W)
I presume 2W is the same as Water Availability (L193). If so, please be consistent with the naming of the indicators, i.e., call an indicator the same all over the manuscript.
Author Response
Rev.#2
Dear Reviewer,
We hereby submit the revised version of the manuscript water-1583330 entitled “Assessing the nexus on local perspective: a quali-quantitative framework for water-energy-food security evaluation in neglected territories”.
After receiving your comments, we thank you very much for providing us the opportunity to further improve our manuscript. We appreciate your constructive criticism and have addressed the major concerns.
In the revised version, all comments were carefully addressed in the text and explained below. If additional changes are needed, we will be happy consider your recommendations. We appreciate your time and look forward to your response.
Sincerely yours,
The Authors
The objectives of the manuscript (research) should be clearly identified, preferably at the end of Introduction.
We rewrote this section.
L189: those seven criteria should be briefly explained.
We added a brief explanation about each criterion.
Access is mentioned in the abstract, L190, Figure 3, Figure 4, etc. But it seems that for water, there is no indicator about access (L193). Probably, it is defined that if water is available to people, then it must be accessible. In other words, you considered that it does not make sense to say that water is available, and then state that it is not accessible. This seems logical, but please make it explicit. On the other hand, Access shows up in Table 1. So, things should be streamlined.
We considered “access” in the questionnaire/table 1, question 9; table 2, indicator 1W.
L207: WHO is an agency of UN, so you shouldn’t write “UN or WHO”. Besides, these are huge organizations. Please make their relevant publications or webpages very clear. You can make a list. Give explicit references and not huge organisations.
Thank you. We rewrote and updated these references.
Table 2: Availability of Public Water Supply Services (2W)
I presume 2W is the same as Water Availability (L193). If so, please be consistent with the naming of the indicators, i.e., call an indicator the same all over the manuscript.
No, it’s not the same. Table 2 presents the indicators available in Brazilian official datasets. So, 2W is an indicator. “Water availability” (L193) is regarding the dimension surveyed. As we re-organize the results and discussions, we believe that it is clearer now.

Reviewer 3 Report
Dear Authors,
Please see the below comments which may help you to improve the article:
- Introduction section – Needs to be divided into four paragraphs a) Establish a territory i.e., in the past decade much research has focused on….reviewing articles of previous research, claiming centrality b) Establish a niche i.e. it remains unclear why….indicating a gap, question-raising, counter-claiming c) Occupying the niche – The purpose of the study was to…..outlining purposes, announcing present research, findings d) Indicating the structure of the paper (restrict it to 2 pages)
- Following articles could be helpful
-
- An Internet of Things Approach for Water Efficiency: A Case Study of the Beverage Factory
- Improving Water Efficiency in the Beverage Industry with the Internet of Thing
- 3. Section 4 is missing.
- Segregate results and discussion as separate sections.
- Limitations and recommendation to be highlighted in the conclusion section clearly.
Author Response
Dear Reviewer,
We hereby submit the revised version of the manuscript water-1583330 entitled “Assessing the nexus on local perspective: a quali-quantitative framework for water-energy-food security evaluation in neglected territories”.
After receiving your comments, we thank you very much for providing us the opportunity to further improve our manuscript. We appreciate your constructive criticism and have addressed the major concerns.
In the revised version, all comments were carefully addressed in the text and explained below. If additional changes are needed, we will be happy consider your recommendations. We appreciate your time and look forward to your response.
Sincerely yours,
The Authors
Dear Authors,
Please see the below comments which may help you to improve the article:
- Introduction section – Needs to be divided into four paragraphs a) Establish a territory i.e., in the past decade much research has focused on….reviewing articles of previous research, claiming centrality b) Establish a niche i.e. it remains unclear why….indicating a gap, question-raising, counter-claiming c) Occupying the niche – The purpose of the study was to…..outlining purposes, announcing present research, findings d) Indicating the structure of the paper (restrict it to 2 pages)
We tried to reduce the introduction and make it clearer.
- Following articles could be helpful
- An Internet of Things Approach for Water Efficiency: A Case Study of the Beverage Factory
- Improving Water Efficiency in the Beverage Industry with the Internet of Thing
Thank you for these inspiring papers.
- 3. Section 4 is missing.
We corrected it.
- Segregate results and discussion as separate sections.
We would like to keep the results and discussions together, to make easier the results understanding.
- Limitations and recommendation to be highlighted in the conclusion section clearly.
We rewrote this section.

Round 2
Reviewer 1 Report
Ms. Ref. No.: water-1583330
Title: Assessing the nexus on local perspective: a quali-quantitative framework for water-energy-food security evaluation in neglected territories
I found this study interesting. In general, the paper now has improved. I think it is suitable for publication with minor language corrections.